# Single pulse all-optical toggle switching of magnetization without gadolinium in the ferrimagnet $Mn_2Ru_xGa$

C. Banerjee[1], N. Teichert[1], K. E. Siewierska[1], Z. Gercsi[1], G. Y. P. Atcheson[1], P. Stamenov [1], K. Rode[1], J. M. D. Coey [1] & J. Besbas[1✉]

Energy-efficient control of magnetization without the help of a magnetic field is a key goal of spintronics. Purely heat-induced single-pulse all-optical toggle switching has been demonstrated, but so far only in Gd-based amorphous ferrimagnet films. In this work, we demonstrate toggle switching in films of the half-metallic ferrimagnetic Heusler alloys $Mn_2Ru_xGa$, which have two crystallographically-inequivalent Mn sublattices. Moreover, we observe the switching at room temperature in samples that are immune to external magnetic fields in excess of 1 T, provided they exhibit a compensation point above room temperature. Observation of the effect in compensated ferrimagnets without Gd challenges our understanding of all-optical switching. The dynamic behavior indicates that $Mn_2Ru_xGa$ switches in 2 ps or less. Our findings widen the basis for fast optical switching of magnetization and break new ground for engineered materials that can be used for nonvolatile ultrafast switches using ultrashort pulses of light.

[1] CRANN, AMBER and School of Physics, Trinity College Dublin, Dublin 2, Ireland. ✉email: besbasj@tcd.ie

Driven by the demands for high speed, low cost, and high-density magnetic recording, research in spintronics has always sought insight into new classes of magnetic materials and devices that show efficient and reproducible magnetization switching. In this respect, interest in the magnetic properties of antiferromagnetically coupled sublattice systems has gained momentum in the last decade. The total or partial cancellation of the sublattice magnetizations makes these systems insensitive to stray magnetic fields, and the interaction between the sublattice spin moments introduces phenomena that are absent in conventional ferromagnets, opening new opportunities for magnetic recording and information processing[1–4].

An efficient way of controlling magnetism is to use ultrashort laser pulses[5–9]. X-ray magnetic circular dichroism (XMCD) investigations in 2011 by Radu et al.[10] of the dynamics of the Gd and Fe atomic moments in a thin layer of amorphous ferrimagnetic $Gd_{0.25}Fe_{0.656}Co_{0.094}$ after a 50 fs laser pulse, revealed a transient parallel alignment of the moments that was the precursor of switching. This was soon followed by the discovery of single-pulse all-optical toggle switching of the magnetization in the same material by Ostler et al.[11]. At that time, a general basis for fast all-optical switching in multisublattice magnets was proposed by Mentink et al.[12]. Amorphous $Gd_x(Fe,Co)_{1-x}$ with $x \approx 25$ is a metallic ferrimagnet with localized $4f$-shell magnetic moments on the Gd sublattice and delocalized $3d$-band moments on the Fe–Co sublattice. Upon excitation by a 10–100 femtosecond laser pulse, the Fe–Co undergoes sub-picosecond demagnetization leading to practically complete loss of the ordered $3d$ shell magnetization, an effect that had been originally observed in ferromagnetic nickel[13]. Concomitantly, the Gd atoms experience a slower loss of magnetic alignment, with partial transfer of angular momentum from the Gd $f$ shell to the Fe–Co $d$ shell[10], entailing a transient parallel alignment of the moments of the demagnetizing Gd and the remagnetizing FeCo that ultimately leads to magnetization toggle switching on a picosecond timescale[10,11]. As the suggested mechanism for single pulse all-optical switching (SP-AOS) relies on an ultrafast interplay between two inequivalent spin sublattices, one with a slower response to the laser (the Gd $4f$ electron shell) and the other with a faster one (the Fe $3d$ electrons), subsequent researches on SP-AOS concentrated on rare-earth based ferrimagnets[14]. In these measurements, it is useful to distinguish between the very short timescale, ~1 ps, on which the future direction of the magnetization is determined, and the longer timescale, ~100 ps, needed for the magnetization reversal to be established when the sample has eventually cooled down.

In practice, helicity-independent SP-AOS has only been demonstrated in ferrimagnetic $Gd_x(Fe,Co)_{1-x}$ thin films[11], $Gd_x(Fe,Co)_{1-x}$ spin valves[8] and in synthetic Gd/Co ferrimagnets[6]. It has not been seen in other rare-earth based ferrimagnets such as amorphous $Tb_{0.27}Co_{0.73}$[15,16], where the $4f$ electrons experience strong spin–orbit coupling. Its thermal origin is established by the independence of the effect on the polarization and helicity of the light[10,11], and the equivalent effect produced by pulses of hot electrons[17]. A related phenomenon has been reported in ferrimagnetic $Tb_x(Fe,Co)_{1-x}$, using nanoantennas[18] and in ferromagnetic Pt/Co/Pt structures, when the laser spot size matches that of the ferromagnetic domains[19]. A different type of single-pulse, nonthermal, nontoggle switching has been reported with linearly-polarized light in insulating Co-doped yttrium iron garnet[20].

In this work, we present a new, rare-earth-free, ferrimagnet that exhibits SP-AOS where, according to the prevailing thermodynamic models[12], the two sublattices should not have drastically different response times to laser excitation because they have almost the same atomic moments. We report all-optical toggle switching in the ferrimagnetic Heusler alloys $Mn_2Ru_xGa$ (MRG)[1] where both magnetic sublattices are composed of manganese, and establish MRG as a versatile alternative to $Gd_x(Fe,Co)_{1-x}$ for SP-AOS applications. In MRG, the Mn atoms occupy two inequivalent sublattices at Wyckoff positions $4a$ and $4c$ in the cubic $F\bar{4}3m$ structure (see Supplementary Fig. 1a), with antiferromagnetic intersublattice coupling[1]. At low temperature the magnetization of the Mn($4c$) sublattice is dominant, but as temperature increases it falls faster than that of the Mn($4a$) sublattice, leading to a compensation temperature $T_{comp}$ where the two are equal and opposite. The coercivity tends to diverge when the net magnetization crosses zero[21]. The value of $T_{comp}$ can be varied by changing the deposition conditions and the Ru concentration $x$, so it is possible to make MRG insensitive to external magnetic fields by decreasing its magnetization[22]. The electronic structures of the two sublattices are different; both have a spin gap of about 1 eV close to the Fermi energy $E_F$, which led to the identification of MRG as the first example of a ferrimagnetic half-metal[1], but the Mn($4c$) electrons have a high, spin-polarized, density of states at $E_F$ whereas that of the Mn($4a$) electrons is much lower. The unusual electronic structure accounts for an anomalous Hall effect (AHE) that is greater than those seen in common ferromagnets[22] and the strong magneto-optical Kerr effect (MOKE) observed even when the net magnetization vanishes at $T_{comp}$[22,23]. Both AHE and MOKE probe mainly the spin-polarized conduction band associated with Mn in the $4c$ position (see Supplementary Notes 2, 3). Domains can be observed directly in the Kerr microscope, regardless of the net magnetization[24].

## Results

**Observation of all-optical toggle switching.** In our experiments, we investigated SP-AOS in 19 MRG thin films having different Ru contents with $T_{comp}$ above or below room temperature (RT). The films are deposited on MgO (100) substrates, which leads to a slight tetragonal distortion of the cubic XA-type structure from space group 216, $F\bar{4}3m$ to space group 119, $I\bar{4}m2$, which is responsible for the perpendicular magnetic anisotropy of the MRG films. Optical pulses of 800 nm wavelength and about 200 fs duration were generated by a modelocked Ti-sapphire laser seeding a 1 kHz amplifier. Figure 1 displays the results of irradiating a $Mn_2Ru_{1.0}Ga$ film by a single 200 fs pulse with a Gaussian intensity profile, as observed by ex situ Kerr microscopy. Here, the light or dark contrast indicates an orientation of the Mn ($4c$) sublattice into or out of the plane. For either initial magnetization direction, a single laser pulse of sufficient intensity switched the magnetization direction in the irradiated area (The elliptical shape of the switched domain is caused by astigmatism of the focusing lens). Pulses, where the average energy density is sub-threshold leave the magnetization unchanged, except at the center, where the intensity may exceed threshold (Fig. 1a). The whole irradiated spot is switched at 1.5 μJ, which corresponds to a laser fluence of 14.5 mJ cm$^{-2}$ (Fig. 1b), but at 29 mJ cm$^{-2}$ a multidomain pattern appears in the center of the irradiated zone (Fig. 1c), where the temperature of the film has transiently approached or exceeded the Curie temperature of the sample (~550 K)[1] leading to remagnetization in submicron domains close in size to the resolution of the Kerr microscope. They are much smaller than the ~100 μm domains normally observed at room temperature after saturating the magnetization[24]. It is established that such temperatures can be reached in equilibrium between the lattice and spin system in the very first picoseconds following optical excitation in transition metal compounds[13], after which the system remagnetizes randomly in the stray field during the cool down. The multidomain pattern is surrounded by a ring-

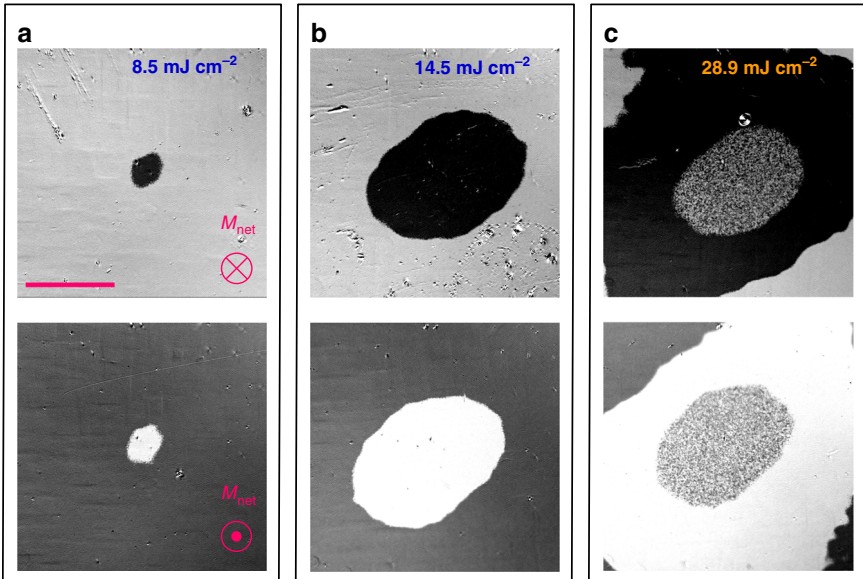

**Fig. 1 Single-pulse all-optical switching (SP-AOS) in Mn$_2$Ru$_{1.0}$Ga.** A uniformly-magnetized film with net magnetization $M_{net}$ out of the plane (top) or into the plane (bottom) is irradiated by a single 800 nm pulse focused onto a ~100 μm spot. The pulse energy in **a** is 8.6 mJ cm$^{-2}$, which only exceeds the switching threshold in a small region at the center, where the fluence is highest. The 14.5 mJ cm$^{-2}$ pulse in **b** switches the whole irradiated area, whereas the 28.9 mJ cm$^{-2}$ pulse in **c** heats the film at the center of the spot above the Curie temperature, producing a fine multidomain pattern. The most intense pulses (43.4 mJ cm$^{-2}$) lead to ablation of the film. The scale bar represents a length of 50 μm.

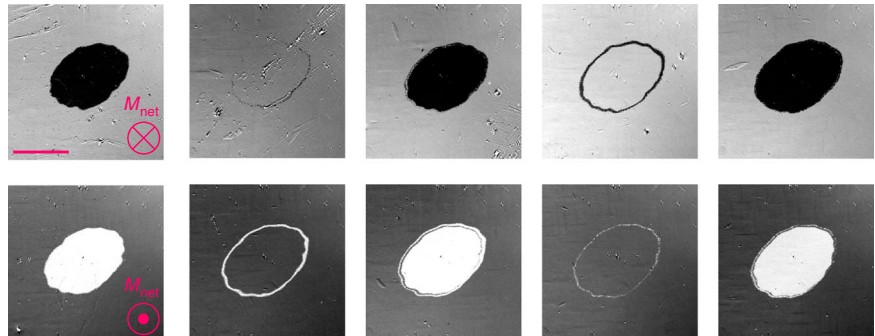

**Fig. 2 Toggling of the magnetization in Mn$_2$Ru$_{1.0}$Ga.** Magnetization patterns are shown as a function of the number of applied pulses. Pulse energy was 11.6 mJ cm$^{-2}$. The scale bar represents a length of 50 μm.

shaped switched region, which shows that SP-AOS involves significant transient demagnetization. The variation of the size of the switched area with increasing pulse energy has been employed to calculate the threshold fluence for switching $F_{th}$ (see Supplementary Note 6) which is found to be ~7.5 mJ cm$^{-2}$. Similarly, the threshold fluence for the formation of a multidomain state is 23 mJ cm$^{-2}$. Interestingly, we never observed SP-AOS in any MRG film having $T_{comp}$ below RT (see Supplementary Note 5 and Supplementary Table 1). We verified that the observed sequence of switching originates solely from laser-induced heating, by repeating the experiments with circularly polarized laser pulses of opposite helicities, different directions of linear polarization with respect to the MRG crystallographic directions as well as using light pulses of 400 nm wavelength (see Supplementary Note 8). The SP-AOS occurred in all cases, which eliminates the possibility of any contribution from magnetic circular dichroism[25] or from transient spin–orbit torques generated by the electric field. On further increasing the laser power to 43 mJ cm$^{-2}$, the centre of the irradiated spot on the film is ablated.

Figure 2 depicts the results of the irradiation with 1–5 successive laser pulses on Mn$_2$Ru$_{1.0}$Ga. The panels show

different regions that were subjected to the given numbers of shots. Consistently, the irradiation by a series of laser pulses leads to a toggling of the direction of the magnetization, which was investigated for up to 12 consecutive pulses.

MRG possesses a low net magnetization and hence a high anisotropy field. Therefore, the coercive field of the films usually exceeds 0.2 T and can reach values as high as 10 T[22] when the temperature is very close to $T_{comp}$. It is interesting to see whether a highly-coercive sample can be switched by light at RT. Figure 3 shows the toggling of magnetization following a sequence of pulses in a film of Mn$_2$Ru$_{0.9}$Ga with coercivity exceeding 1 T. That sample could not be saturated in our electromagnet, so it was measured in its virgin state, which is characterized by a distribution of magnetic domains with a predominance of magnetization directed toward the substrate. Toggling of each individual domain by the light pulse is observed even though the sample is insensitive to an external magnetic field of 1 T. The threshold fluence for this sample was approximately one third of that of Mn$_2$Ru$_{1.0}$Ga. The SP-AOS observed in samples with compensation temperature close to RT is particularly important for three reasons: (1) the threshold fluence required for switching

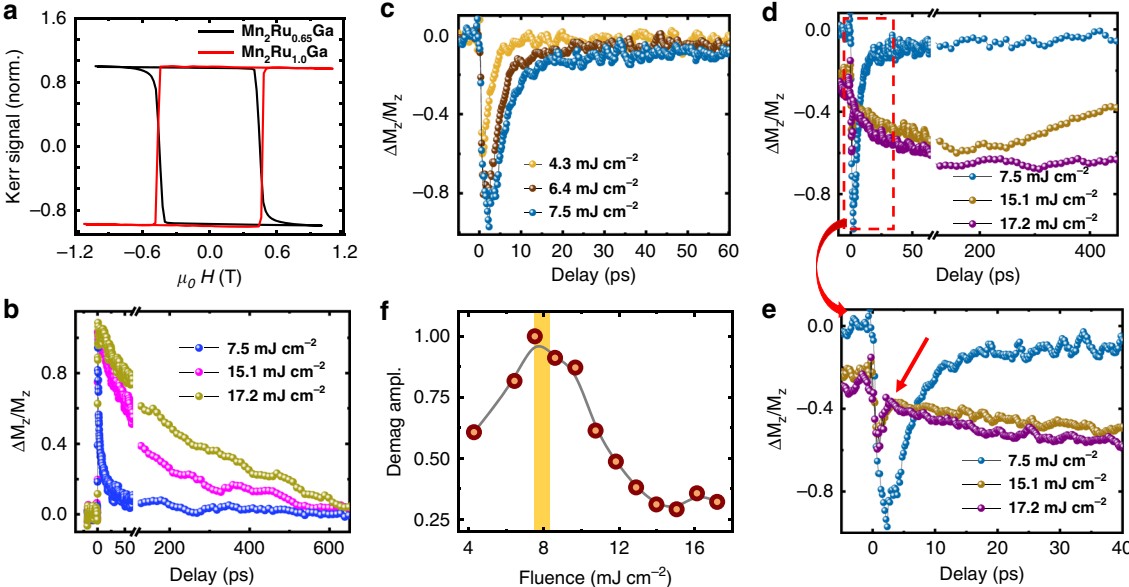

**Fig. 3 Toggling of magnetization in a high-coercivity Mn₂Ru₀.₉Ga film.** Repeated toggling of the micron-scale domain pattern of a virgin-state sample is observed with repeated pulses. There was a net imbalance of domains pointing in and out of the plane. The scale bar represents a length of 50 μm.

**Fig. 4 Time resolved magnetization dynamics in Mn₂RuₓGa. a** Hysteresis loops measured by MOKE in Mn₂Ru₀.₆₅Ga and Mn₂Ru₁.₀Ga, which have compensation temperature below and above RT respectively. **b** Transient Kerr signals of Mn₂Ru₀.₆₅Ga for different pump fluences. The variation of the Kerr signal is normalized to the total Kerr rotation at RT. **c** Transient Kerr signal of Mn₂Ru₁.₀Ga for fluences below the switching threshold; **d** similar data including fluences above threshold; **e** A zoom of **d** in a shorter time window; the anomaly marked by the arrow is discussed in the text. **f** Variation of the demagnetization amplitude at zero delay for different pump fluences. The yellow shaded region indicates the threshold fluence for switching. The solid line is a guide to eye.

is small, reducing accumulated heat and potentially enabling energy-efficient applications in the future[26]; (2) the coercivity of MRG diverges close to $T_{comp}$, which makes the magnetic state insensitive to external magnetic fields; (3) switching of micron-sized domains is possible, which are much smaller than the laser spot size.

**Ultrafast magnetization dynamics.** Next we turn to the dynamics of the excitation and reversal processes. The magnetism in MRG originates from the 3$d$ moments of the Mn(4$a$) and Mn (4$c$) sublattices, which are antiferromagnetically coupled. After the action of a femtosecond pulse, the free electron temperature rises rapidly through the Curie temperature, reaching values above 1000 K[27] (see also Supplementary Note 9). At this temperature, the interatomic exchange interactions (<0.1 eV) are overcome and the magnetic order is rapidly destroyed; while the intra-atomic, on-site exchange that depends on stronger Coulomb interactions (3–5 eV) should be partially preserved. The aftermath of the pulse therefore involves re-establishment of sublattice magnetic order from the atomic moments, which in a ferrimagnet could include effects of angular momentum transfer between the sublattices. To investigate this possibility, we have studied the magnetization dynamics using time-resolved polar MOKE (TR-MOKE) in the two-color collinear pump/probe geometry. In this part of the study, we compare two samples,

Mn₂Ru₁.₀Ga and Mn₂Ru₀.₆₅Ga. They have $T_{comp}$ of 390 K (see Supplementary Note 4) and 165 K, respectively and their coercive fields at room temperature are similar (~460 mT), as shown in the optically measured hysteresis loops in Fig. 4a. The loops have opposite signs, as expected, because the Mn(4$c$) sublattice, which gives the dominant contribution to the MOKE signal, aligns parallel to the applied field below $T_{comp}$ and antiparallel above. Intense laser pulses of wavelength 800 nm were used as the pump to excite the magnetization dynamics, and the Mn(4$c$) sublattice magnetization was subsequently probed in a stroboscopic manner using weaker 400 nm pulses. A field of 500 mT was applied perpendicular to the films to ensure an identical initial state before each pump pulse.

Figure 4b shows the TR-MOKE signal for different pump fluences for the Mn₂Ru₀.₆₅Ga sample, which does not switch because $T_{comp}$ is below RT. Following the laser excitation, the transient MOKE signal shows a step-like change, caused by the ultrafast destruction of the magnetic order of the Mn(4$c$) sublattice as the electron temperature shoots up. Less than 2 ps after the pulse, the electrons share their energy with the lattice leading to an increase of lattice temperature of 100–200 K[27] (Supplementary Note 9). Subsequently, the magnetization regains its initial state after tens or hundreds of ps, depending on the increase of lattice temperature. The data for the Mn(4$c$) sublattice here resemble those for a ferromagnetic metal such as Ni. It

should be noted that even though the MOKE response indicates full demagnetization of the Mn($4c$) sublattice, a trace of magnetic order could still persist in the magneto-optically silent Mn($4a$) sublattice, which has the greater moment throughout the temperature range above compensation. At greater fluences we observe a multi-domain state after laser irradiation, indicating that the system has been thermally demagnetized when the sublattices re-establish thermal equilibrium.

On changing to $Mn_2Ru_{1.0}Ga$, we find a strong dependence of the TR-MOKE signal on laser fluence, which is quite different below and above the threshold fluence $F_{th}$ for SP-AOS (see Fig. 4c, d respectively). Below threshold, the behavior is like that of $Mn_2Ru_{0.65}Ga$ at similar fluence; the recovery takes about 10 ps, and an increase in the fast demagnetization signal is observed with increasing fluence (Fig. 4f). Upon crossing the fluence threshold indicated by the yellow bar, the following new features appear in the signal: (i) an offset at negative delay, not observed at comparable fluences for the nonswitching sample in Fig. 4b, accompanied by a change of shape and loss of MOKE contrast. This offset may be an effect of the stray field from the magnet imprinting a pattern on the film during the 1 kHz optical irradiation, in a fashion similar to analog magnetic recording with AC bias[28]; the almost binary toggling of the central region within the pump focal spot plays the role of the AC bias field. (ii) An increase in the pump fluence now leads to a decrease in the demagnetization amplitude at zero delay (Fig. 4e, f), contrary to the previous sample (Fig. 4b) and to $Gd_x(FeCo)_{1-x}$[29]. (iii) As the system relaxes, the signal undergoes a rapid partial recovery within 2 ps (see Fig. 4e), after which its slope becomes negative, at the point marked by the arrow. The higher-fluence signals continue like this (Fig. 4d), before being turned back after 50 ps by the weak applied field and gradually recover over the course of several hundred picoseconds. Extrapolating the negative slope to negative saturation of the switchable component gives a switching time, in the longer sense mentioned in the introduction, of about 200 ps. The timescale for recovery of the first sample (Fig. 4b) at high fluence is similar. The features described begin to appear in the time resolved signal after the laser fluence crosses $F_{th}$, (see Supplementary Note 7) and they become prominent at ~14 mJ cm$^{-2}$, where the entire spot area is switched (Fig. 1).

Indications of switching in the dynamics above $F_{th}$ in our data are: first, the drastic change within the first 2 ps suggesting an exchange of angular momentum between the sublattices, which leads to the transient ferromagnetic-like state with parallel orientation of sublattice magnetizations that is generally accepted as necessary for switching[10]. Second, the subsequent negative slope in Fig. 4e that is opposite in sign to the signal below $F_{th}$ at that timescale, which indicates that in absence of magnetic field, the system will relax in the switched state.

Another sample with $T_{comp} = 250$ K, excited at 210 or 230 K was found by Bonfiglio et al. to behave similarly[27] and the time for thermalization of the electrons and the lattice, also deduced there from a 4-temperature model, was 2 ps. Furthermore, they have shown that magnetic order is already beginning to be re-established in MRG within 1 ps, permitting efficient exchange scattering and transfer of angular momentum from one sublattice to the other, even at extremely short timescales.

The key difference between MRG and GdFeCo is that while the latter exhibits SP-AOS when measured within about 100 K either above or below its compensation point[11,30], MRG only switches when the initial temperature is below $T_{comp}$. Therefore, initially, the angular momentum of the 4c sublattice must be higher than that of 4a to switch successfully. This strongly suggests that the driving mechanism for switching in MRG is exchange scattering with conservation of angular momentum within the spin systems, which is in agreement with current understanding of the

exchange-driven part of the switching in GdFeCo[31,32] and recent studies of the switching in MRG with pulses of various durations and wavelengths from a free-electron laser[33]. In order to switch, the sublattice with the stronger intra-sublattice exchange, Mn($4a$) in our case, must cross zero before the weaker. This process can be fast, ~100 fs, but a shorter wavelength of the probe pulse and a better time resolution would be needed to reveal the behavior of the Mn($4a$) sublattice and be quite sure that the transient parallel alignment of the two sublattice moments that is seen in XMCD in $Gd_x(FeCo)_{1-x}$[10] is also present in MRG.

In conclusion, we have demonstrated single-pulse all-optical thermal switching in less than 2 ps in films of the half-metallic compensated Heusler ferrimagnet $Mn_2Ru_xGa$, where both magnetic sublattices are composed of manganese atoms, occupying different crystallographic sites. These results extend the scope of the phenomenon beyond the limited range of amorphous $Gd_x(Fe,Co)_{100-x}$ alloys with $x \approx 25$, where the magnetic sublattices are defined chemically. A comparison of the two systems is provided in Supplementary Note 10 and in Supplementary Table 2. The Heusler alloys are a huge family, with an established body of knowledge about their magnetic and electronic properties that will allow us to advance our understanding of single-pulse all-optical switching and design materials that can be the basis of future nonvolatile opto-magnetic switches. Beyond the newly-demonstrated quality of MRG as an opto-magnetic material, its large intrinsic spin–orbit torque, which relies on the absence of inversion symmetry in the Mn($4c$) sublattice opens prospects for new multifunctionality[34–36]. Therefore, MRG and its chemically tailored successors offer the prospects of both new insights into condensed matter on a femtosecond timescale and new technological prospects that take advantage of ultrafast control of the magnetic state without any reliance on a magnetic field.

## Methods

**Sample preparation**. MRG films with different Ru content were grown on MgO (001) substrates at 350 °C by DC magnetron sputtering in a Shamrock system with a base pressure of $2 \times 10^{-8}$ Torr. They were co-sputtered from Ru and stoichiometric $Mn_2Ga$ targets. The Ru concentration was controlled by varying the $Mn_2Ga$ target plasma power while fixing the Ru power. The samples were then capped with a protective layer of 2 nm of $Al_2O_3$.

**MOKE microscopy**. Femtosecond laser pulses were generated by Ti-sapphire laser seeding a 1 kHz amplifier with a Q-switched cavity. Their central wavelength was 800 nm and the pulse duration was about 200 fs. The amplifier can be operated in continuous mode, where a train of pulses is generated at a repetition rate of 1 kHz or in single pulse mode where the emission of one single pulse can be externally triggered. In some cases, 400 nm laser pulses were obtained by second harmonic generation in a β-BaB$_2$O$_4$ crystal.

Prior to laser irradiation, the films were saturated at room temperature in the 1 T perpendicular magnetic field of an Evico Kerr microscope. Different areas of the films were then irradiated with several linearly polarized laser pulses of different powers, followed by ex-situ imaging of the results in the Kerr microscope. For the imaging, a polarized beam is focused onto the sample using a microscope objective. A rotation in the polarization due to Kerr effect occurs in the reflected beam, which then passes through an analyzer, before reaching the camera. In order to increase the contrast, the axis of the analyzer is kept few degrees away from the cross position in both directions to acquire two images. Subsequently, the difference is taken to extract the final image.

**TR-MOKE measurements**. For the dynamic measurements, the laser beam with wavelength 800 nm was split into a pump beam and a frequency-doubled probe beam at 400 nm. The intensity of the probe was kept low. Both pump and probe were linearly polarized and collinear. The spot sizes were measured to be about 150 and 70 μm, respectively. The dynamical magneto-optic Kerr rotation was measured using a balanced photo detection scheme and acquired using a lock-in amplifier and a mechanical chopper at 500 Hz in the pump beam. In order to extract the magnetic signal, the transient magnetic response was acquired for two directions of the magnetic field, which were subsequently subtracted from each other. The pump/probe delay was varied using a mechanical delay line.

## Data availability

The authors declare that the data supporting the findings of this study are available within the paper and its supplementary information files.

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

## Acknowledgements

This project has received funding from Science Foundation Ireland through contracts 16/IA/4534 ZEMS and 12/RC/2278 AMBER and from the European Union's FET-Open research programme under grant agreement No 737038. C.B. is grateful to Irish Research Council for her post-doctoral fellowship. N. T. would like to acknowledge funding from the European Union's Horizon 2020 research and innovation programme under the Marie Skłodowska-Curie EDGE grant agreement No 713567. The authors thank Prof. John Donegan for his valuable comments on the work.

## Author contributions

C.B., J.B., and J.M.D.C. designed the project. Experimental work was done by C.B., N.T., and J.B. Growth and characterization of the samples were carried out by G.A. and K.S. Numerical simulations were performed by P.S., Z.G., and J.B. C.B., J.B., P.S., J.M.D.C., and K.R. interpreted the data. All authors discussed the results. C.B., J.B., K.R., and J.M.D.C. wrote the paper.

## Competing interests

The authors declare no competing interests.
