## [Peer Review File · Nature Communications]

Reviewers' Comments:

Reviewer #1:

Remarks to the Author:

I still believe that the discovery of a second material class showing single-pulse all-optical switching is of extreme interest not only to the ultrafast spin dynamics community but also to those working in spintronics, and why not for those working in topological physics in Heusler alloys.

Please read previous own reports for more detailed arguments supporting this statement.

This is why I strongly recommend its publication in *Nature Communications*.

Yet, from a technical point of view, it is non-evident from the time resolved MOKE data that switching occurs at ultrafast time scales.

As I suggested already in my previous report, it would be great that the experimental results would be backed up by a simpler theory or picture than the involved discussion provided by the authors.

However, from my point of view, this is not a major drawback since the authors have been able to engineer a completely new class of material practically from scratch (see their previous publications) with excellent properties for spintronics which (accidentally?) also shows single-pulse all-optical switching.

One may argue that in the original work of Ostler et al. where for the first time SP-AOS using heat alone was demonstrated, the authors presented an excellent and very complete work, where experiments in thin films as well as on nanostructures were conducted, supported by atomistic spin dynamics simulations. Thus, in comparison the present work seems to be incomplete for some reviewers. However, one should point out that at the time, year 2012, several works were already published on the amorphous GdFeCo ferrimagnet regarding their magneto-optical properties and its reaction to femtoseconds pulses including switching, such as multi-pulse all-optical switching (Stanciu et al. PRL 2007) and single-pulse switching (I. Radu et al. *Nature* 2011), to name a few. Even nowadays, almost every week a new work is published that claims to shed new light into the switching mechanisms in GdFeCo, or to demonstrate its potential for spintronics applications, etc... In view of the results of the current work I foresee a great success and impact to MRG in the field of AOS and spintronics.

Some thoughts I would like to share with the authors:

Related to Ref. 28: There, similar experiments were conducted including some of the authors of the current work. Indeed, in that work MRG with $x=0.7$ was studied at ambient temperature close but still below the compensation point. So, in view of the results in the present work, conditions for switching in Ref. 28 should be optimal. I guess that in Ref. 28 the laser fluence ($6.5\text{mJ}/\text{cm}^2$) was not sufficient to fully demagnetize the Mn 4c sublattice, so switching is not observed, or at least switching is not claimed there. At the same time, one would expect that the dynamics observed in Ref. 28, which corresponds to $x=0.7$, and $F=6.5\text{mJ}/\text{cm}^2$, would be similar to that presented in this work for $x=0.65$ and $F=7.5\text{mJ}/\text{cm}^2$ (for example). However, the dynamics that are observed in Ref. 28 are surprisingly very similar to that presented in Figures 4c. In Ref. 28, the anomaly appearing in Figure 4c at 2ps was convincingly explained as a result of very different dynamics of the 4a and 4c spin energy (or temperature). While Mn 4c reacts almost instantaneously to the laser heat input due to the low spin specific heat, the Mn 4a presents a much slower dynamics due to the 6 times larger spin heat

capacity. Since the sublattices are strongly exchange coupled, after these 2 ps where Mn 4a and Mn 4c temperature is different, they reach the same spin temperature and continue demagnetizing at the rate determined by Mn 4a, which in turn is demagnetised due to the energy/angular momentum exchange to Mn 4c.

This means that at least initially, in the first 2 ps, it looks like Mn 4a is way slower than Mn 4c. This also means that the exchange of angular momentum is stronger/faster/larger than in GdFeCo, as it is already observed in fluence regimes where switching does not occur. If this were right, ultrafast switching would very likely in situations where Mn 4c is fully demagnetize while Mn 4a magnetization remains finite. I would say this is exactly what is going on in the current work. If the authors agree with me, I would suggest them to rephrase the text were they suggest that the magnetization dynamics of each sublattice is similar, in contrast to GdFeCo. I believe, in agreement with Ref. 28, that one sublattice is slow (Mn 4a) and the other is fast (Mn 4c). In this case, the mechanism for switching would be the same as in GdFeCo but just more efficient in a better material for spintronics.

At the same time, something that puzzles me is that in Ref. 28, the anomaly in the magnetisation dynamics happens for non-switching laser fluences, while in the present work, this anomaly is only observed for fluences above the critical fluence for switching. It seems like there exists a characteristic energy in the spin system that should be activated to start transferring energy from one sublattice to the other. I am just guessing but I would say that this energy should scale somehow with the compensation temperature, as this temperature is critical the onset for switching, and in Ref. 28, the experiments were conducted very close to the compensation temperature.

Reviewer #2:

Remarks to the Author:

The experimental demonstration of the toggle switching is convincing and novel and could trigger further research on this material and on all-optical switching. I do understand that not all experimental data can be explained right away, and in some cases the experimental demonstration is important by itself. In response to the referees comments, the authors attempted to provide the qualitative explanation to the experimental data, which, in my opinion, needs to be improved, if the manuscript is to be considered in Nature Communications. In the present form there are several confusing statements, which would harm the accessibility of the manuscript to a reader. Therefore, the manuscript cannot be recommended for publication in the present form.

1. The data shown in Fig. 4 are important for understanding the switching scenario suggested by the authors. Therefore, it is surprising that in Fig. 4c2 no curves are shown for the fluences between the switching threshold F_{th} and the fluence sufficient for emergence of the multidomain state. How does the slower dynamics after 2 ps evolve with increase of the fluence? The arguments of the authors regarding the signature of switching at 2 ps would be more clear and convincing, if they would show the dynamics at e.g. 9 mJ/cm², 10 mJ/cm² etc.

2. The authors provide only rough estimations (order of magnitude) of the electrons and lattice temperature increase and do not specify for which laser fluences these temperatures are given. It is not clear for me, why the authors do not provide the calculated numbers based on material parameters given in Ref. 28. As a result, it is unclear, is there any correlation between F_{th} and a fluence required to heat the lattice above T_{comp} , for instance. Can it have an impact on the switching process? The authors do not discuss this issue at all when they describe the data in Fig. 4. At the same time, in the manuscript they refer to the paper [29] where a very similar dynamics was discussed in relation to the switching across the compensation point. Although I fully understand and appreciate that not all experimental findings can be explained right away, somewhat more accurate

discussion of possible factors affecting the dynamics would greatly increase a possible impact of the paper.

3. In page 6 the authors provide the number for the lattice temperature increase (100-200 K). To which particular fluence these values are related?

4. In the main text the authors describe that they have derived the threshold fluence and refer to the supplementary material, but do not specify F_{th} . This value should be given in the main text. Further, it would be helpful if the fluence at which the multidomain state emerges would be also given and indicated at Fig. 4d.

5. I do not really understand the sentence in page 7: "Extrapolating the initial slope...". What does "...switching time, in a longer sense,.." means? Possibly, showing the extrapolated curve in Fig. 4 would help to make it more clear for a reader. In the same sentence, the authors discuss the timescale for recovery. Could the authors be more specific here? What is the conclusion from this observation?

6. Technical question. How were the curves shown in Fig. 4 obtained? Was the transient polarization rotation measured at positive and negative field, and the curves were subtracted from each other? This is a standard approach to exclude non-magnetic contribution to the measured signal. This is not discussed in methods section.

7. It is somewhat confusing that in a part of the manuscript the laser pulse is characterized by energy [μJ], and in some – but the fluence [mJ/cm^2].

8. In the introduction the authors state that the switching in TbFeCo has been reported under specific structural conditions. To the best of my knowledge, current understanding is that the switching in TbFeCo is similar to the one in GdFeCo, but it is challenging to stabilize the switched state. Therefore, I think it would be more fair to state in the introduction that the toggle switching was observed in both GdFeCo and TbFeCo.

9. Ref. 3 cited in the abstract does not report on the toggle switching. To the best of my knowledge, the toggle switching was clearly demonstrated for the first time in Ref. 12.

Response Letter

Manuscript Title: Single-pulse all-optical toggle switching of magnetization without Gd: The example of $\text{Mn}_2\text{Ru}_x\text{Ga}$.

Manuscript ID: NCOMMS-20-20355-T

Authors: C. Banerjee *et. al.*

Journal Name: Nature Communications.

The reviewers' feedback show that they read our work carefully. Their comments have contributed to the further improvement of the manuscript and we are grateful for that.

The interest shown by all the reviewers in our work is highly motivating. We appreciated the pertinent comment, "up to date it has been impossible to find a new class of material showing thermal single pulse switching despite intensive research effort made by the community", that recognizes the intensive work of present and past team members to develop MRG. That reviewer recommended publication in Nature Materials. We also appreciated the comment "demonstration of the toggle single-pulse switching in a new material is of high importance and interest", which is an opinion clearly shared by all reviewers.

In the following, we address both reviewers' comments, including those relating to the qualitative explanation of the observed magnetization dynamics. We trust the manuscript is now suitable for publication in Nature Communications.

Reviewer 1

I still believe that the discovery of a second material class showing single-pulse all-optical switching is of extreme interest not only to the ultrafast spin dynamics community but also to those working in spintronics, and why not for those working in topological physics in Heusler alloys.

Please read previous own reports for more detailed arguments supporting this statement.

This is why I strongly recommend its publication in Nature Communications.

Yet, from a technical point of view, it is non-evident from the time resolved MOKE data that switching occurs at ultrafast time scales.

As I suggested already in my previous report, it would be great that the experimental results would be backed up by a simpler theory or picture than the involved discussion provided by the authors.

However, from my point of view, this is not a major drawback since the authors have been able to engineer a completely new class of material practically from scratch (see their previous publications) with excellent properties for spintronics which (accidentally?) also shows single-pulse all-optical switching.

One may argue that in the original work of Ostler et al. where for the first time SP-AOS using heat alone was demonstrated, the authors presented an excellent and very complete work, where experiments in thin films as well as on nanostructures were conducted, supported by atomistic spin dynamics simulations. Thus, in comparison the present work seems to be incomplete for some reviewers. However, one should point out that at the time, year 2012, several works were already published on the amorphous GdFeCo ferrimagnet regarding their magneto-optical properties and its reaction to femtoseconds pulses including switching, such as multi-pulse all-optical switching (Stanciu et al. PRL 2007) and single-pulse switching (I. Radu et al. Nature 2011), to name a few. Even nowadays, almost every week a new work is published that claims to shed new light into the switching mechanisms in GdFeCo, or to demonstrate its potential for spintronics applications, etc... In view of the results of the current work I foresee a great success and impact to MRG in the field of AOS and spintronics.

Response:

We appreciate this evaluation of our work.

Our aim is to be able to publish the information on the discovery of a new class of materials that exhibit single-pulse all-optical switching. It is not to present another theory of the phenomenon. We are happy to discuss briefly and clearly our ideas about what is going on, which we do on p. 8 of new revised version.

One of the main observations we make in MRG is that SP-AOS only occurs when the compensation temperature is above room temperature, the starting point for all our measurements. Therefore, the equilibrium angular momentum of the 4c sublattice must initially be higher than that of the 4a sublattice. This strongly suggests that the driving mechanism for switching is exchange scattering with conservation of angular momentum within the spin systems. This

relaxation path can be fast, typically the associated timescale is ~ 100 fs. Based on this, we believe that the normal spin-lattice relaxation mechanism that dominates switching in GdFeCo, which unlike MRG switches when measured within about 100 K *above* or below T_{comp} , is almost absent in MRG. This hypothesis entails that the transient ferromagnetic-like state (the necessary precursor for switching) is reached in less than a picosecond. In our data, this leads to a change of slope in the demagnetisation curves at around $t \sim 0.5$ ps. Switching, in the sense that the system will necessarily relax later towards the opposite magnetic polarity, therefore occurs before ~ 2 ps. We also note that this is in agreement with the current understanding of the exchange-driven part of switching in GdFeCo [see for example U. Atxitia et. al. Physical Review B 89, 224421 (2014)] – In order to switch, the sublattice with the stronger intra-sublattice exchange Fe (in our case 4a) must cross zero before the weaker one Gd (in our case 4c).

As regards the referee's further thoughts in relation to Ref 28, the data there were collected some time ago, before we had discovered toggle switching. Both from the fluence and the dynamics, we are confident that the sample studied there does undergoes SP-AOS. The fluence is sufficient, in view of the closeness of the initial temperature to the compensation point. The similarity with the dynamics with those of the sample we measured in the present work above threshold is striking. Below threshold the dynamics look quite different.

We appreciate the referee sharing their thoughts on the process. Our own are slightly different, as outlined above.

Response to Reviewer 2

Comment #1: The experimental demonstration of the toggle switching is convincing and novel and could trigger further research on this material and on all-optical switching. I do understand that not all experimental data can be explained right away, and in some cases the experimental demonstration is important by itself. In response to the referees' comments, the authors attempted to provide the qualitative explanation to the experimental data, which, in my opinion, needs to be improved, if the manuscript is to be considered in Nature Communications.

Response: We have presented our current understanding of the data, as summarized above, on page 8 at the end of the paper.

Comment #2: The data shown in Fig. 4 are important for understanding the switching scenario suggested by the authors. Therefore, it is surprising that in Fig. 4c2 no curves are shown for the fluences between the switching threshold F_{th} and the fluence sufficient for emergence of the multidomain state. How does the slower dynamics after 2 ps evolve with increase of the fluence? The arguments of the authors regarding the signature of switching at 2 ps would be more clear and convincing, if they would show the dynamics at e.g. 9 mJ/cm², 10 mJ/cm² etc.

Response: We have a complete set of data at nine different fluences above threshold, as could be seen from Fig 4d. We only show a few of them in Fig 4c2 and Fig4c3 to avoid cluttering the figure and to focus on the fluences, 15.1 and 17.2 mJ/cm², where the entire irradiated area has switched. We have included three more intermediate curves at lower fluences in the Supplemental Information VII, as the referee requests. There, six curves are plotted separately and the evolution after 2 ps with the increase of fluence can be examined.

Comment #3: The authors provide only rough estimations (order of magnitude) of the electrons and lattice temperature increase and do not specify for which laser fluences these temperatures are given. It is not clear for me, why the authors do not provide the calculated numbers based on material parameters given in Ref. 28. As a result, it is unclear, is there any correlation between F_{th} and a fluence required to heat the lattice above T_{comp} , for instance. Can it have an impact on the switching process? The authors do not discuss this issue at all when they describe the data in Fig. 4. At the same time, in the manuscript they refer to the paper [29] where a very similar dynamics was discussed in relation to the switching across the compensation point. Although I fully understand and appreciate that not all experimental findings can be explained right away, somewhat more accurate discussion of possible factors affecting the dynamics would greatly increase a possible impact of the paper.

Response: In our view the only necessary condition for MRG to switch that the 4c sublattice is initially the one with the greater angular momentum. The lattice would only reach T_{comp} after the transient ferromagnetic-like state has decayed. For information, we show calculations in the Supplementary Material IX, based on the 4T parameterization, of transient effective lattice, electron and spin temperatures using the parameters given in Ref. 28. The modelling confirms that laser heating at threshold fluence does indeed raise the lattice to T_{comp} after about 2 ps.

As regards ref 29 where the authors switch GdFeCo. We agree that the negative slope after switching is similar to our observation, but do not agree that the dynamics are very similar; there seems to be an order of magnitude difference in the timescales involved.

Comment #4: In page 6 the authors provide the number for the lattice temperature increase (100-200 K). To which particular fluence these values are related?

Response: We now include the details of the lattice temperature increase for different laser fluences in the Supplementary Information IX. This is now mentioned on p.6.

Comment #5: In the main text the authors describe that they have derived the threshold fluence and refer to the supplementary material, but do not specify F_{th} . This value should be given in the main text. Further, it would be helpful if the fluence at which the multidomain state emerges would be also given and indicated at Fig. 4d.

Response: These values are now added in the main text on page 4.

Comment #6: I do not really understand the sentence in page 7: “Extrapolating the initial slope...”. What does “...switching time, in a longer sense,..” means? Possibly, showing the extrapolated curve in Fig. 4 would help to make it more clear for a reader. In the same sentence, the authors discuss the timescale for recovery. Could the authors be more specific here? What is the conclusion from this observation?

Response: The negative slope referred to is the slope after 2ps of the two higher-fluence curves in Fig 4c3. We distinguish in the introduction (bottom of p.2) between (i) switching on a (sub)picosecond timescale when the transient ferromagnetic-like state that evolves into an embryonic switched configuration is established, and (ii) the magnetization reversal that is established when the sample has eventually cooled down, a process that takes hundreds of picoseconds.

Comment #7: Technical question. How were the curves shown in Fig. 4 obtained? Was the transient polarization rotation measured at positive and negative field, and the curves were subtracted from each other? This is a standard approach to exclude non-magnetic contribution to the measured signal. This is not discussed in methods section.

Response: The data were indeed obtained using the standard approach by subtracting the transient MOKE signal for two directions of the magnetic field. We have now added a sentence in the Methods section to this effect.

Comment #8: It is somewhat confusing that the in a part of the manuscript the laser pulse is characterized by energy [uJ], and in some – but the fluence [mJ/cm²].

Response: We have now included the fluences corresponding to the pulse energies in the first part of the manuscript.

Comment #9: In the introduction the authors state that the switching in TbFeCo has been reported under specific structural conditions. To the best of my knowledge, current understanding is that the switching in TbFeCo is similar to the one in GdFeCo, but it is challenging to stabilize the switched state. Therefore, I think it would be more fair to state in the introduction that the toggle switching was observed in both GdFeCo and TbFeCo.

Response: We have now been specific, stating on p.2 that the switching in TbFeCo was observed using nanoantennas .

Comment #10: Ref. 3 cited in the abstract does not report on the toggle switching. To the best of my knowledge, the toggle switching was clearly demonstrated for the first time in Ref. 12.

Response: This was a mistake, which we have now corrected by interchanging refs 3 and 12.

Summary of Changes Made in the Revised Manuscript:

- 1) The references 3 and 12 are now interchanged. The references 29, 31, 32, 33 are added.
- 2) In the first paragraph of page 3, the sentence starting with “A related phenomenon has been.....” is modified.
- 3) In page 4, the laser fluences are added.
- 4) In page 5, the first four lines are modified to add the threshold fluence for switching as well as for formation of multidomain state.
- 5) The text in page 7 and 8 is modified starting from “On changing to Mn₂Ru_{1.0}Ga, we find.....” upto the paragraph before conclusion.
- 6) In page 10, one sentence is added to the last paragraph in Methods section.

- 7) In Fig. 1, the pulse energies are converted to laser fluence and the figure caption is changed accordingly.
- 8) In the Supplementary Information, sections VII and IX are added, the numbering of the other sections is changed accordingly.

Reviewers' Comments:

Reviewer #1:

Remarks to the Author:

As I have already discussed in previous reports, the authors have discovered/designed a new class of material for ultrafast heat-induced switching. This new class of material seems to be ideal for many technological applications. This work presents this discovery with enough data to support it. The very details of the microscopic origin of the switching process are left open for the research community. The publication of this work fits perfectly the goals of a journal such as Nature Communication, therefore I strongly recommend its publication in its present form.

Reviewer #2:

Remarks to the Author:

In response to the referee's comments, the authors have clarified their interpretation of the switching mechanism, and also improved presentation of some data.

Although the switching scenario suggested by the authors needs to be confirmed in more comprehensive experiments and by theoretical consideration, the importance of the experimental results reported in the manuscript justifies publication in Nature Communications.

Manuscript Title: Single-pulse all-optical toggle switching of magnetization without Gd: The example of $\text{Mn}_2\text{Ru}_x\text{Ga}$.

Manuscript ID: NCOMMS-20-20355A

Authors: C. Banerjee *et. al.*

Journal Name: Nature Communications.

We thank all four reviewers for their comments that helped improving the quality of this work. No further comments are left to address. The present file is provided for completeness.

Reviewer #1 (Remarks to the Author):

As I have already discussed in previous reports, the authors have discovered/designed a new class of material for ultrafast heat-induced switching. This new class of material seems to be ideal for many technological applications. This work presents this discovery with enough data to support it. The very details of the microscopic origin of the switching process are left open for the research community. The publication of this work fits perfectly the goals of a journal such as Nature Communication, therefore I strongly recommend its publication in its present form.

Reviewer #2 (Remarks to the Author):

In response to the referee's comments, the authors have clarified their interpretation of the switching mechanism, and also improved presentation of some data.

Although the switching scenario suggested by the authors needs to be confirmed in more comprehensive experiments and by theoretical consideration, the importance of the experimental results reported in the manuscript justifies publication in Nature Communications.